# Population, distribution, biomass, and economic value of Equids in Ethiopia

**Girma Birhan Asteraye**[1,2,3,4]*, **Gina Pinchbeck**[2], **Theodore Knight-Jones**[1,3], **Klara Saville**[5], **Wudu Temesgen**[1,3,4], **Alemayehu Hailemariam**[6], **Jonathan Rushton**[1,2]

**1** Global Burden of Animal Diseases Programme https://animalhealthmetrics.org, **2** Institute of Infection, Veterinary and Ecological Sciences, University of Liverpool, Neston, United Kingdom, **3** International Livestock Research Institute, Addis Ababa, Ethiopia, **4** College of Veterinary Medicine and Animal Sciences, University of Gondar, Gondar, Ethiopia, **5** The Brooke, London, United Kingdom, **6** Brooke Ethiopia, Addis Ababa, Ethiopia

* Girma.Asteraye@liverpool.ac.uk, G.birhan@cgiar.org

## Abstract

### Background

Equids play a crucial role in the Ethiopian economy, transporting agricultural inputs and out-puts in the dominant subsistence agricultural systems and the critical link for value chains throughout the country. However, these species are often neglected in policies and interventions, which reflects the data and information gaps, particularly the contribution of working equids to Ethiopia.

### Objective

To assess population dynamics, distribution, biomass, and economic value of equids in Ethiopia.

### Materials and methods

Equine population data were obtained from the Ethiopian Central Statistics Agency (CSA) annual national agriculture surveys published yearbooks from 2004 to 2020. Parameters such as the number of effective service days and daily rental value were obtained from interviews and literature to estimate the stock monetary and service value of equids. Descriptive statistics were used to assess population dynamics and the geographical distribution was mapped.

### Results

The estimated total Ethiopian equid population increased by more than doubled (by 131%) between 2004 and 2020 from 5.7 (4.9–6.6) million to 13.3 (11.6–15) million with 2.1 million horses, 10.7 million donkeys, and 380 thousand mules. Similarly, the number of households owning a working equid has increased. Equine populations are unevenly distributed across Ethiopia, although data were lacking in some districts of the country. The per human-capita equine population ranged from 0–0.52, 0–0.13, and 0–0.02 for donkeys, horses, and mules, respectively. The equid biomass was 7.4 (6.3–8.4) million Tropical livestock unit (TLU) (250

**Funding:** GBA would like to acknowledge Brooke and Horse Trust for their funding this study. The funders had no role in study design, data collection and analysis, decision to publish, or preparation of the manuscript.

**Competing interests:** The authors have declared that no competing interests exist.

kg liveweight), 10% of total livestock biomass of the country. The stock monetary value of equids was USD 1,229 (651–1,908) million, accounting for 3.1% of total livestock monetary value and the services value of equids was USD 1,198 (825–1,516) million, which is 1.2% of Ethiopian 2021 expected GDP.

## Conclusion

The Ethiopian equine population has grown steadily over the last two decades. Equids play a central role in transportation and subsistence agriculture in Ethiopia and contribute significantly to the national economy. This pivotal role is insufficiently recognized in national livestock investments.

## Introduction

Equids are important for farming and non-farming households playing a key role in the national economy of Ethiopia [1], being a widely used mode of transport and draught power [2]. They also have important cultural value, being prominent in festivals and entertainment, with ceremonial roles in religious services such as weddings and funerals [2–5].

Working equids (horses, donkeys, and mules) play a significant role in generating household income [2, 6, 7]. For instance, Admassu and Shiferaw [3], demonstrated that equids provide 14% of annual household income, which accounted for 227 USD, which was comparable to other livestock species. In some parts of Ethiopia, the money gained from each working animal can support between five and twenty family members [2]. In a study around Addis Ababa households given a donkey by a local NGO reported an improvement in income and 84% of female donkey owners indicate that the donkey improved their lives [8]. In addition, owning a working animal is a major source of empowerment for women in terms of being freed from daily subsistence chores, and their wider status among the community [6]. In peri-urban and urban areas of Ethiopia working animals are used, mainly by youths, to generate an income by transporting merchandise (3,9,10). Furthermore, the gross value of equine transportation and draft services in Ethiopia was estimated at Ethiopian Birr 18,959 million [1] and 7,035 million [9] in 2010. Working equids were recognized by the UN as working livestock in 2016 and considered 'critical to the livelihoods and resilience of millions of families throughout low- and middle-income countries' [10].

Despite their considerable socioeconomic importance, equids are often neglected in livestock programs and investments [2, 3]. This leaves working equids vulnerable to low welfare standards and suboptimal output [2, 11, 12].

Ethiopia has the second largest working equid population after Mexico, with 40% of sub-Saharan Africa's horses and donkeys and over 90% of the subcontinent's mule population [13, 14] and several NGOs are working in Ethiopia to improve the lives of working equids [2].

A number of studies have highlighted the problems of obtaining accurate population statistics on working animals and the challenges in assessing their economic value [14–16]. However, in Ethiopia much of the research up to now has been limited to the prevalence of some diseases and assessments of welfare [17, 18].

Since 1990/91 the Central Statistical Agency (CSA) of Ethiopia has carried out surveys and a solitary agricultural census and reported the annual population size for different livestock species [19]. Similarly, from 1961 to the present, FAOSTAT statistical data repository, provides free access concerning food and agricultural data from around the world [14].

Although the FAOSTAT database is an easily available source for understanding the size and distribution animals, it does not categorize sub-populations, such as by age or purpose. In addition, the FAOSTAT figures appear to be underestimates, particularly in Africa [17, 20]. Thus, we have used all the available published annual production yearbooks on livestock and livestock characteristics obtained from the CSA [19]. This study describes the population dynamics, distribution, biomass, and economic value of equids in Ethiopia. Understanding the size, distribution, and economic value of working equids will allow more informed decisions, particularly concerning investments in livestock development.

The Global burden of animal disease (GBADs) is a program, launched in 2018 [21–23], aiming measuring and understanding the global burden of animal diseases. GBADs has been working on estimation of biomass and economic value across all livestock species to have common denominators to estimate direct and indirect losses caused by animal diseases worldwide. The present study is a part of this program and the findings with biomass and economic value of the equids could be used as initial data for further analysis of the burden of animal disease in working equids in Ethiopia. Moreover, the result includes important findings relevant to study the competition for grazing and the balance between natural resource use and the generation of sustainable activities for Ethiopia.

## Methodology

### Data sources and collection

Information on the size of Ethiopian livestock populations was obtained from CSA (https://www.statsethiopia.gov.et/our-survey-reports/) annual report series, the agricultural sample survey from 2004 to 2020. We used all available published yearbooks. In addition, donkey, horse, and mule population data from the FAOSTAT [13] website from 2004 to 2019 were used to show the trend difference compared to CSA.

CSA estimates the livestock data based on the information obtained from the livestock holders within the sampled agricultural households in the entire rural areas of the country, including all pastoralist areas. However, due to data sampling constraints and sometimes due to insecurities, livestock population data were not reported in some areas in 2020/21. The CSA data cover all domestic animals such as cattle, sheep, goat, camel, equine, and chickens by age, sex, and the purpose for which they are raised, such as transport, draught, and other uses in case of equids [19].

The price of a live animal (cattle, sheep, goats, and camels) was taken from the livestock market information system database of the Ministry of Trade and Industry (MoTI) of Ethiopia for the year 2021 [24]. Because MoTI does not report equids market prices in the system, seventeen (7 knowledgeable elders and 10 livestock market brokers) were interviewed for the current average market price and number of service days for transportation, from 17 districts where Brooke (an international NGO) operates to protect and maintain the health and welfare of equids in Oromia, Amhara, and Southern Nations Nationalities and Peoples (SNNPR) regions. These are dominant areas where donkeys, horses, and mules are used for transportation and draught power. These data are collected by local Brooke representatives and cluster team leaders after receiving oral consent from respondents. One interviewee per district was selected purposively based on their knowledge of use frequency and equids' market price. In the interview, we asked about the average current market price of equids, and the average service days worked by equids each week, regardless of the number of hours worked per day. We then calculated the average market price and the average number of workdays. The interview guide is provided in S1 File.

The rental value of equids for transportation, as well as rental value and number of service days for draft usage, were obtained from a study done by Metaferia et al [9]. By considering the inflation rate in the last decade as indicated in the formula below,

$$RV2021 = RV2011 + (RV2011 * IR2011) + \cdots + RV2020 + (RV2020 * IR2020)$$

Where, RV = Rental value, IR = Average inflation rate of the year.

The reference rental value in 2011 was taken from Metaferia et al [9], while the country's average inflation rates were obtained from CSA annual reports.

## Data processing, analysis, and mapping

Stock monetary value was calculated by multiplying the total number of equid population by the current average market price in the year 2020. Equid service value was also determined by multiplying the total number of working equids (aged 3 years and above) by their average number of service days and rental value in 2020. Similarly, Biomass was calculated by using 250 kg as one Tropical livestock unit (1 TLU) [25] and multiplying it by the total equid population number.

The CSA [19] data disaggregated horses, donkeys and mules aged 3 years and above according to their services, mainly as transportation, draught, and others, and this categorization was used to estimate biomass, stock monetary, and service value. Working equids used for transporting goods or people by pulling carts or packing as transportation by the CSA, whilst working equids used for agricultural ploughing is categorized as draught and breeding equids as others. In addition, the existing livestock production system classifications of the Ministry of Agriculture (MoAG) of Ethiopia, Crop-livestock mixed (CLM) and Pastoral, as well as the list of administrative zones grouped under these production systems, were used to show the monetary value and biomass of equids in different production systems. The S2 File contains a complete dataset, including all information.

The geographical distribution of equids population was mapped using zone-level number of equids per capita (since the lower scale livestock data available are zone level), calculated as equids population divided by the human population, with human population data obtained from CSA [19]. Quantum GIS 3.16.11 software (QGIS Geographic Information System. Open-Source Geospatial Foundation Project. http://qgis.osgeo.org) was used to map equid populations.

The national and regional equine population dynamics were entered into Microsoft Excel. Simple descriptive statistics with tabulations and graphs were employed to show the trends. The equid population percentage change was calculated by dividing the difference between the initial equine population in 2004 and the population in 2020 by the initial population and multiplying by 100. Sampling uncertainty was expressed by calculating a 95% confidence interval (CI) from CSA equids population estimates and standard errors. Sensitivity analysis was checked by ± 20% to assess how individual input parameters influenced the output service value of equids.

## Results

### Equine population dynamics and service days

The total Ethiopian equid population has increased by more than double (131%) between 2004 and 2020, from 5.7 (4.9–6.6) million to 13.3 (11.6–15) million. CSA data showed roughly steady growth in the equine population between the year 2004 and 2020 (Fig 1).

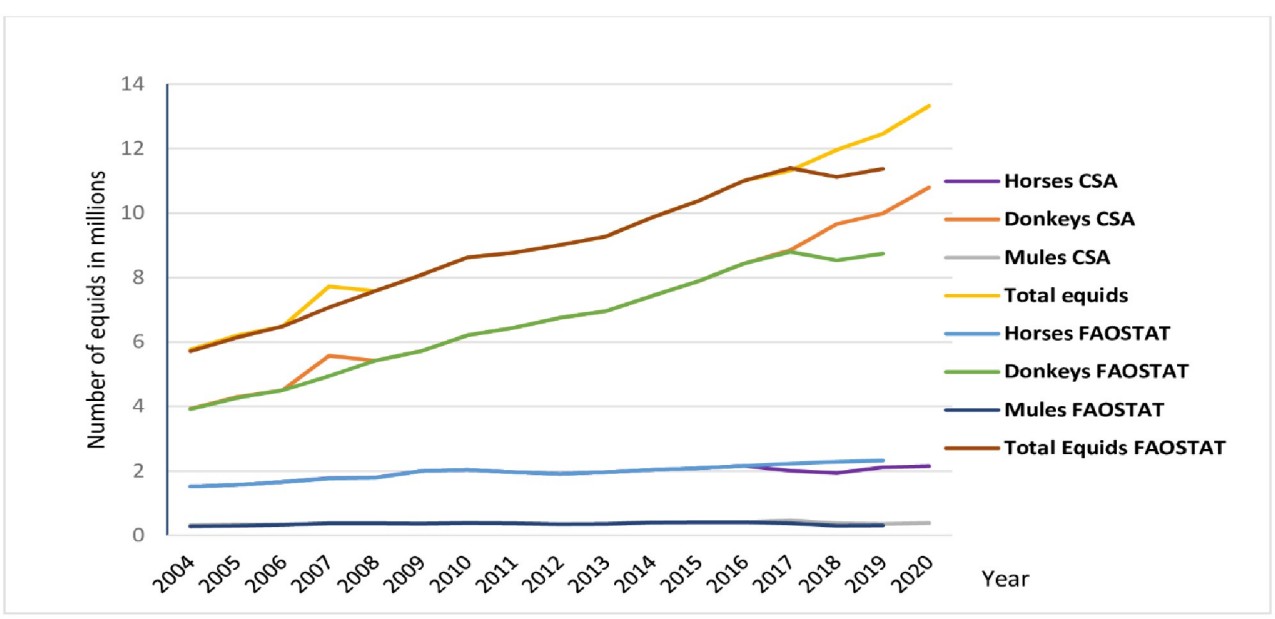

**Fig 1. Ethiopian equid population dynamics.**

The largest growth of the equid population was recorded in 2006 (6.2 million to 6.4 million, increase by 19.2%) and the largest reduction was recorded in 2008 (7.7 million to 7.5 million, decrease by 1.8%) (Fig 1). The Oromia region has the largest share 40% of equine population in Ethiopia, followed by Amhara and SNNPR regions, which have 33.2% and 9.1% of equids, respectively (Table 1).

The increase in the population was also observed in each individual species with the highest increase observed in donkeys. Between 2004 and 2020, the donkey population increased by 24 and 9 fold in Afar and Somali pastoral areas, respectively, with a high percentage of change was observed starting from 2018 (Table 1).

Despite an overall increase in the number of horses and mules in the country. The reduction in the number of mules was documented in some reginal states (Table 1). CSA does not report the population number of both horses and mules in Afar, Somali, Harari, and Dire Dawa regions.

CSA data suggests a steady growth in the number of households that own working equid, with an increase of 154% between 2004 and 2020. The number of donkey holders showed the highest growth and was followed by horse holders with the value of 52% and 23.7% respectively. Between 2004 and 2020, the proportion of donkey owners increased from 25.1% to 38.1%. the proportion of horses and mules' owners, on the other hand, are decreasing (Table 2).

According to the interviewees, the effective service days for a donkey, horse, and mule are approximately 2.98 (2–4), 1.95 (1.3–2.6), and 1.89 (1.2–2.6) days per week on average for transportation, resulting in 155 (104–205), 101 (68–135), and 98 (62–135) days of service per year for donkeys, horses, and mules, respectively. Similarly, the market price for donkeys, horses and mules was estimated to be 74.8 (47.6–98), 151.8 (95.2–211.6), and 250 (148–333) USD, respectively. Because the data is insufficient, we did not estimate the effective service days of equids for draught usage.

**Table 1. The change in number of donkeys, horse, and mule population in different regions of Ethiopia.**

| Regions | Donkey Population (Thousand Head) (Range) | | | Horse Population (Thousand Head) (Range) | | | Mule Population (Thousand Head) (Range) | | |
|---|---|---|---|---|---|---|---|---|---|
| | **2004** | **2020** | **Change (%)** | **2004** | **2020** | **Change (%)** | **2004** | **2020** | **Change (%)** |
| Tigray | 386.6 (292.2–481) | 901 (690–1,112) | 133 | 1.2 (0.1–2) | 2.6 (0.08–5) | 113 | 9.2 (0.4–18) | 28.1 (15–41) | 210 |
| Afar | 12.3 (3.2–21.3) | 308.8 (121–497) | 240 | - | - | - | 0.2 (0.06–0.3) | - | - |
| Amhara | 1,400 (1,208–1,592) | 3,725.4 (3,163–4,287) | 167 | 257 (102–412) | 490.2 (202–778) | 90.5 | 89 (45.5–133) | 199.8 (99–300) | 125 |
| Oromia | 1,704 (1,427–1,981) | 3,901 (3,263–4,539) | 129 | 959.7 (561–1,359) | 1,309.9 (794–1,826) | 36.5 | 153.7 (95.7–212) | 120.2 (25–215) | - 22 |
| Somali | 91.5 (49–134) | 978.8 (775–1,182) | 97 | - | - | - | - | 0.54 (0.06–1) | - |
| B/Gumuz | 37.5 (19.3–56) | 62.6 (28.2–97) | 67 | - | - | - | 1.8 (0.09–4) | 3.5 (0.6–7) | 95 |
| SNNPR | 278.4 (186–371) | 818.6 (617–1,021) | 194 | 298.7 (146–451) | 345 (198–492) | 15.5 | 63.5 (38–89) | 45.5 (18–73) | - 28 |
| Gambela | - | 0.6 (0.1–1) | - | - | - | - | - | - | - |
| Harari | 6.3 (1.4–11) | 14.6 (12–17.5) | - | - | - | - | - | - | - |
| Dire Dawa | 88 (4.3–11.4) | 24.7 (2–47) | - | - | - | - | - | - | - |
| **National** | 3,930 (3,566–4,295) | 10,792 (9,847–11,737) | 175 | 1,5176 (1,063–1,971) | 2,148 (1,535–2,762) | 41 | 317.6 (240–395) | 382.5 (240–525) | 20 |

key: (-) No data\, (B/Gumuz) Benishangul Gumuz, (SNNPR) Southern nations nationalities and peoples region.

## Equid population distribution

The populations of donkeys, horses, and mules are unevenly distributed across Ethiopia. The majority of equids were reported in highland regions of the country, but still, considerable numbers of donkeys were found in pastoral areas. The per capita equine population ranges from 0 to 0.52 for donkeys, 0 to 0.13 for horses and 0 to 0.02 for mules in different administrative zones of the country (Figs 2–4). Generally, higher per capita of equine populations were observed in the zones grouped under crop livestock mixed system of the country where agricultural activities are prevalent.

## Biomass and stock monetary value of equids

The average equine biomass was estimated to be 7.4 (6.3–8.4) million TLU, accounting for 10% of Ethiopia's total livestock biomass (Table 3) (S1a Fig), and the total stock monetary value of equids was estimated to be 1,229 (651–1,908) million USD, accounting for 3.01% of Ethiopia's total livestock economic value (Table 3) (S1b Fig). The total population, stock

**Table 2. The proportion of working equid-holding households in total livestock-holding rural households in 2004 and 2020.**

| Equine holdings | Year 2004 | | Year 2020 | |
|---|---|---|---|---|
| | Number | Percent | Number | Percent |
| Horse holdings | 968,088 | 8.6% | 1,470,534 | 7.8% |
| Donkey holdings | 2,830,734 | 25% | 7,200,897 | 38% |
| Mule holdings | 295,594 | 2.6% | 365,684 | 1.9% |
| Total equids holdings | 4,094,416 | 36.2% | 9,037,115 | 47.9% |
| Total livestock holdings | 11,296,840 | | 18,877,022 | |

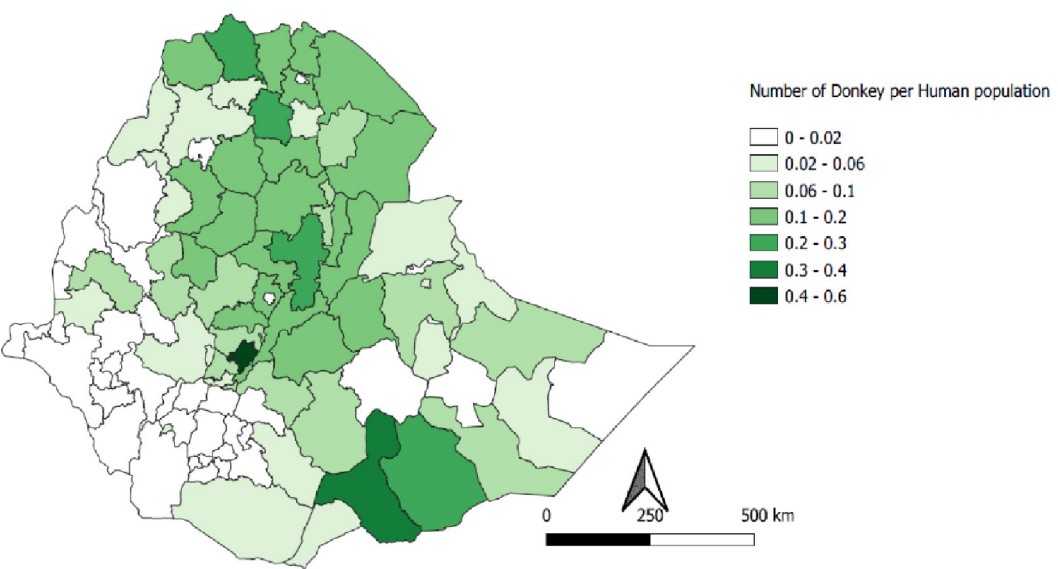

**Fig 2. Per capita distribution of donkeys.**

biomass and monetary value of different equine species, production system and their purpose are presented in Table 4.

## Transportation and draft services of equids

The service value of equine transportation and draft was estimated to be 1,197.6 (825–1,516) million USD. Most of the service value of equine transportation and draft was created by donkeys. Values by species and purpose of equids is presented in Table 5. The estimated service

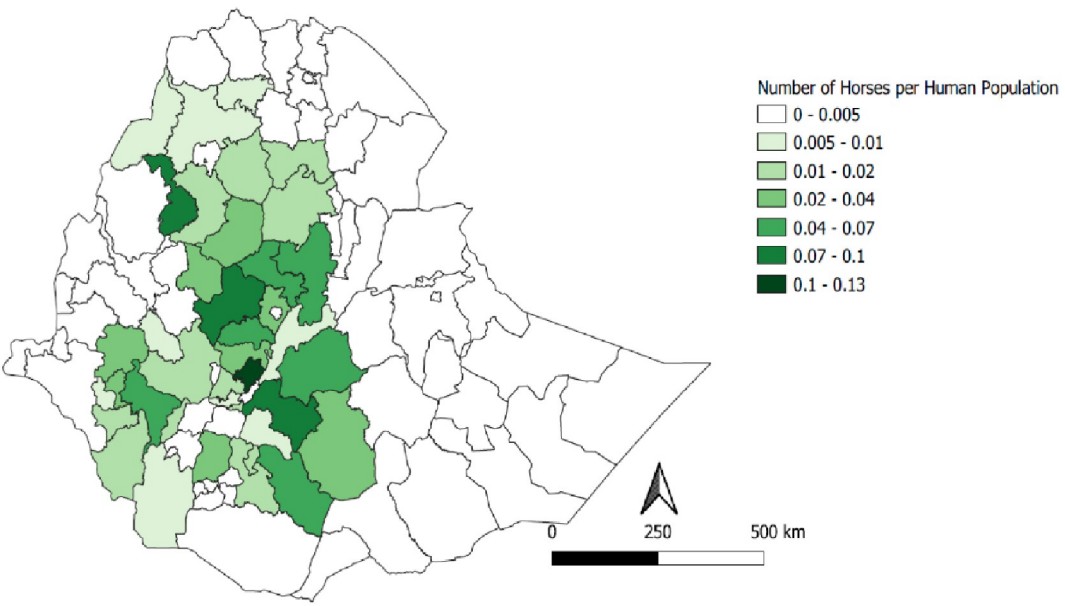

**Fig 3. Per capita distribution of horses.**

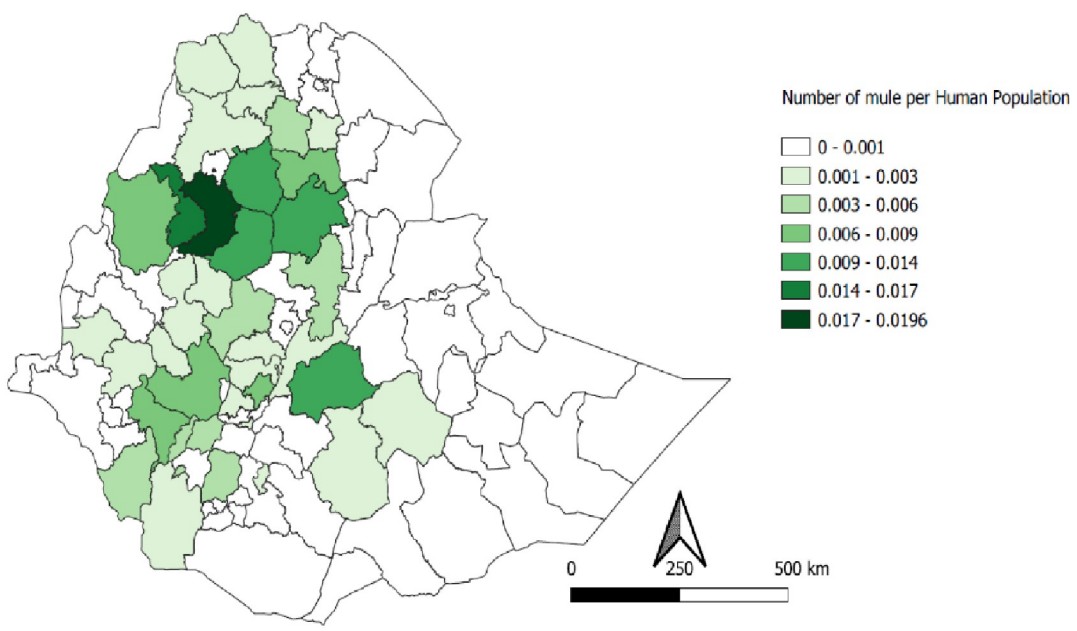

**Fig 4. Per capita distribution of mules.**

value of equids accounts for up to 1.2% of the 2021 year expected national GDP of the country. Sensitivity analysis was checked for each input variable for estimation of service value and the result found sensitive mainly for the change in population of equids.

## Discussion and conclusions

Ethiopia has a large equine population and play a central role in Ethiopia's agricultural and transport systems. It could be said that working equids, particularly donkeys, are part of the critical infrastructure of Ethiopia e.g., essential for the functioning of a society and the economy [1, 3, 6].

Over the last fifteen years (2004–2020), the equid population has grown by more than double, this is primarily due to increases in the number of holdings, which may be related to the increase in human population. For instance, in rural Ethiopia, where 85% of the population of the country lives, when a new family is established, most begin living by owning a donkey as a working animal for transporting both farm and non-farm items because of poor road-network development. However, this could be due to an increase in awareness of owners to report the presence of large numbers of donkeys as livestock to the survey; because donkeys are not food animals in Ethiopia, their owners may not have previously reported them as livestock assets in

**Table 3. The equids population number, biomass, and the monetary value.**

| Species | Population (Head in million) (Range) | Biomass (Million TLU) (Range) | Stock monetary value (USD in million) (Range) |
|---|---|---|---|
| **Donkey** | 10.8 (9.8–11.7) | 5.4 (4.9–5.7) | 807.3 (469–1,149) |
| **Horse** | 2.1 (1.5–2.8) | 1.7 (1.2–2.2) | 326.2 (146–584) |
| **Mule** | 0.38 (0.2–0.5) | 0.3 (0.17–0.5) | 95.6 (36–175) |
| **Total equids** | **13.3 (11.6–15)** | **7.4 (6.3–8.4)** | **1,229.1 (651–1,908)** |

**Table 4. The number of equids population, their stock monetary value and biomass in the different production systems and purpose.**

| Species | Classification | Category | Population (Head in million) | Biomass (Million TLU) | Stock monetary value (USD in million) (Range) |
|---|---|---|---|---|---|
| **Donkey** | Production system | CLM | 8.9 | 4.4 | 664(423–868) |
| | | Pastoral | 1.4 | 0.7 | 106(67–139) |
| | Purpose | Transport | 7.4 | 3.7 | 556(354–727) |
| | | Draught | 0.8 | 0.4 | 60 (38–78) |
| | | Other use | 0.2 | 0.1 | 14(9–18) |
| **Horse** | Production system | CLM | 2 | 1.6 | 302(189–420) |
| | | Pastoral | 0.2 | 0.2 | 28(18–39) |
| | Purpose | Transport | 1.4 | 1.1 | 210 (132–293) |
| | | Draught | 0.3 | 0.2 | 41 (26–58) |
| | | Other use | 0.1 | 0.1 | 14 (9–20) |
| **Mule** | Production system | CLM | 0.4 | 0.2 | 86 (51–115) |
| | | Pastoral | 0.01 | 0.01 | 2 (1–3) |
| | Purpose | Transport | 0.3 | 0.2 | 73 (43–97) |
| | | Draught | 0.1 | 0.03 | 12(7–16) |
| | | Other use | 0.01 | 0.01 | 3 (1–3) |

**Note:** The slight difference in the population numbers between Tables 3 and 4 arises from the process of classification by production system and purpose.

the survey in the same way that cattle, sheep, or goats are, and this is supported by Starkey and Starkey [20].

The Brooke's "Invisible Helpers" report [26] also highlighted fuel price rises have been one of the major drivers of the increased equine population alongside growing human populations and climate change. The increase in number of equids in the present study is consistent with the global trends that Norris et al [14] found between 1997 and 2018 for an increase in equid populations and demonstrates that this trend is continuing. In certain situations, CSA reports contradict some equid population growth trends [14, 20]. For instance, in 2006 an unexpected surge of equids population growth rate (19.2%, which is questionable) was reported. This might be due to the artifact in the CSA data collection or summarization process.

There is an increase in working equids ownership, primarily with donkeys. The donkey population make up the overwhelming majority of the equid population. This indicates the continued significant contribution of donkeys to individual households, and better resilience in different production systems. This is supported by Starkey & Starkey's finding suggesting an increase in the number and demand for donkeys in sub-Saharan African countries [20]. CSA

**Table 5. Estimates of service value of equids.**

| Equine species | Services of Equines | Population | Service Days/year | Rental services (USD/day) | Value (Million USD/year) |
|---|---|---|---|---|---|
| **Donkey** | Transportation | 7,430,131 | 155 (104–205) | 0.8 | 913.8 (613.2–1,208.6) |
| | Draft (pair) | 397,731 | 48 | 1.6 | 30.3 |
| **Horse** | Transportation | 1,383,012 | 104 (68–135) | 1.3 | 183.3 (123.4–244.9) |
| | Draft (pair) | 135,829 | 42 | 3.7 | 21.1 |
| **Mule** | Transportation | 290,067 | 104 (62–135) | 1.6 | 45.1 (28.5–62.1) |
| | Draft (pair) | 23,197 | 42 | 4.2 | 4.1 |
| **Services Value** | | | | | 1,197.6 (824.9–1515.6) |

expand the geographic coverage of the survey area by including three zones of Afar region as of 2018 and six zones in Somali region as of 2019 which had not been included in the previous surveys. This suggests that the CSA data consistently underestimates the donkey population figure in previous years because CSA surveys have been confined to sedentary farming zones, excluding pastoral areas. This also results in a difference in the trend of equid population in the last three years between the CSA and FAOSTAT report.

Authors expected a reduction in the number of donkeys due to global demand for donkey skin, which is used to make traditional medicine *ejiao* [27], and reports of illegal cross-border donkey trade to Kenya [7]; however, Ethiopia subsequently closed the established donkey slaughterhouses before they began operation and banned donkey slaughtering following public affront for the business perceived as offensive to societal values and norms of the country, and while control of illegal donkey smuggling has begun, the impact on the donkey population in Ethiopia is not visible or perceptible in the CSA report thus far.

With regards to the horses and mules though, the global mule and horse population appears to be in a steady decline with a 64% and 0.6% reduction over the past twenty years, respectively [15]. The current study showed modest growth with some fluctuations in the numbers of horses and mules demonstrating the country still relies heavily on equids for transport and traction. Despite their overall population stability, the current study revealed the reduction in mules and horses in some regional states over time. This could be due to variation in market price between the neighbouring regions and potential movement and trade of mules and horses between the regions. Notably, the reason for the reduction of mules in some regions might be due to mules no longer being considered as prestigious animals in the community compared with previous times. As well, the importation of male breeding donkeys ("*Sinar*") from Sudan in the west to central Ethiopia is currently difficult due to political instability in the area. This could lead to shortage of male breeding donkeys which in turn influences a number of mules.

Equid populations are unevenly distributed in Ethiopia and a higher equid population is found under the crop livestock mixed system. This could be due to the CLM practiced in the highlands, which covers about 40% of the country's land mass, and the dominant crop producing area; farmers require equids to transport their agricultural inputs and outputs in those areas. Similarly, the equids count per capita was higher in this system. However, the trend of equid population mainly donkeys showed remarkable growth in pastoral areas. This result is highly supported Wilson [28], stated as donkeys are second in water conservation mechanisms next to the camel among domestic animals, they can survive well in drought prone areas, and they are affordable as compared to camels.

The number of service days of equids varies depending on the production system and the owners' living setup; for example, equids raised in rural and crop-producing areas are heavily involved in activities such as agricultural production and transportation.

The estimated equine biomass accounts for about 10% of the total livestock biomass of the country, contributing almost as much biomass as small ruminants and camels. This estimate may also be used as a denominator for various livestock-related analyses and species comparisons, including feed requirements, greenhouse gas emissions, and other environmental consequences like deforestation and land degradation.

The equines service value contributes about 1.2% of the 2021 year expected national GDP of the country. This indicates the considerable services of equids to the national economy mainly in delivering services for transportation and a source of draught power. The present service value estimate is double to the corresponding gross output estimated by Behnke's [1] and almost five times the result by Metaferia et al. [9]. This could be mainly due to the growth in population of equids, the inflation in service value in the last decades, the difference in

methodologies employed and the underestimation of equids working days per year in previous studies. The current study used part of the data and methodology of Metaferia et al. [9], who employed the UN System of National Accounts (UN-SNA) to estimate gross value added in livestock operations (or livestock GDP). However, we did not deduct intermediate consumption from the gross value of working equid output because most of the equine population lives in a rural, extensive system, and most equids rely on natural resources that still need to be appropriately valued. Even Metaferia et al. [9] deducted other livestock's total intermediate consumption from the gross value of working equid output, which is questionable. Behnke et al. [1] used a production approach for estimating gross outputs. However, the technique they used to arrive at the figures is unclear. Moreover, their research used data from one administrative zone of Ethiopia, which may not be representative of the entire country. Since the economic role of equids are not yet studied systematically in Ethiopia the present estimated results are based on a rudimentary set of information, and it may have a degree of error. This is a very important area for future research. We recommend a review to improve estimation of livestock contribution to the national GDP of Ethiopia.

Despite the broad range of data and information that can be gained from the CSA data sets, there are limitations with the information on cart horses and mules and packing donkeys which are known to exist in urban and peri-urban areas, as no organized data is available for this sector nationally and this data set covers only the rural household-based holdings. Because of the above, our estimated stock monetary and service value lacks the economic contribution of carthorses, cart mule and packing donkeys in urban and peri-urban areas of the country which play a critical role in these local economies. Furthermore, there are no sources elucidating a production or husbandry classification system for working equids and their main production systems and environments under which they are maintained are needed. This kind of study will create the foundation for NGOs and policy makers to understand where and which equids are in the greatest need.

## Supporting information

**S1 File. Interview guide for current market price, number of services days and rental value.**
(DOCX)

**S2 File. The dataset.** Which comprises equine and human populations at the zone level, the number of equids population in different production systems, equine populations by species, age and purpose, the number of equines holding rural households, market price and biomass.
(XLSX)

**S1 Fig. The proportion of equids biomass (a) and stock monetary value (b) with total livestock biomass and stock monetary value.**
(DOCX)

## Acknowledgments

The authors acknowledge all cluster team leaders of Brooke Ethiopia for their assistance in equid market value and service day data collection. The authors also thank Brooke for co-supervising.

## Author Contributions

**Conceptualization:** Girma Birhan Asteraye, Gina Pinchbeck, Theodore Knight-Jones, Klara Saville, Wudu Temesgen, Jonathan Rushton.

**Data curation:** Girma Birhan Asteraye, Alemayehu Hailemariam.

**Formal analysis:** Girma Birhan Asteraye, Wudu Temesgen.

**Investigation:** Girma Birhan Asteraye, Gina Pinchbeck, Theodore Knight-Jones, Klara Saville, Wudu Temesgen, Jonathan Rushton.

**Methodology:** Girma Birhan Asteraye.

**Supervision:** Gina Pinchbeck, Theodore Knight-Jones, Klara Saville, Wudu Temesgen, Alemayehu Hailemariam, Jonathan Rushton.

**Visualization:** Girma Birhan Asteraye.

**Writing – original draft:** Girma Birhan Asteraye.

**Writing – review & editing:** Girma Birhan Asteraye, Gina Pinchbeck, Theodore Knight-Jones, Klara Saville, Wudu Temesgen, Jonathan Rushton.

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
