## [Decision Letter · Decision Letter 0]

9 Mar 2023

PONE-D-23-03880Population, distribution, biomass, and economic value of equids in EthiopiaPLOS ONE

Dear Dr. Asteraye,

Thank you for submitting your manuscript to PLOS ONE. After careful consideration, we feel that it has merit but does not fully meet PLOS ONE’s publication criteria as it currently stands. Therefore, we invite you to submit a revised version of the manuscript that addresses the points raised during the review process.

We look forward to receiving your revised manuscript.

Kind regards,

Chisoni Mumba

Academic Editor

PLOS ONE

3. We note that you have referenced (Aynalem BG. Seroprevalence, Associated risk factor and economic impact of African Horse sickness in Gondar zone, northwest Ethiopia. Unpublished MSc Thesis, University of Gondar. 2021.) which has currently not yet been accepted for publication. Please remove this from your References and amend this to state in the body of your manuscript: (ie “Bewick et al. [Unpublished]”) as detailed online in our guide for authors

4. We note that Figure 4 in your submission contain [map/satellite] images which may be copyrighted. All PLOS content is published under the Creative Commons Attribution License (CC BY 4.0), which means that the manuscript, images, and Supporting Information files will be freely available online, and any third party is permitted to access, download, copy, distribute, and use these materials in any way, even commercially, with proper attribution. For these reasons, we cannot publish previously copyrighted maps or satellite images created using proprietary data, such as Google software (Google Maps, Street View, and Earth). For more information, see our copyright guidelines: http://journals.plos.org/plosone/s/licenses-and-copyright.

a. You may seek permission from the original copyright holder of Figure 4 to publish the content specifically under the CC BY 4.0 license. 

**Comments to the Author**

1. Is the manuscript technically sound, and do the data support the conclusions?

Reviewer #1: Yes

Reviewer #2: Partly

2. Has the statistical analysis been performed appropriately and rigorously? 

Reviewer #1: N/A

Reviewer #2: No

3. Have the authors made all data underlying the findings in their manuscript fully available?

Reviewer #1: No

Reviewer #2: No

4. Is the manuscript presented in an intelligible fashion and written in standard English?

Reviewer #1: Yes

Reviewer #2: Yes

5. Review Comments to the Author

Reviewer #1: Many thanks to the authors for submitting this interesting paper. I have three main comments and some smaller points made by line at the end of this review

- My main question reading this paper is why the results are not presented as ranges. The CSA data sampling methodology are included, and therefore a confidence interval or similar could be calculated. In a similar way, there is insufficient information about how other values are calculated, e.g. from key informant intervals. Presenting these interim figures, as well as the final figures, as ranges, would help the reader understand the range that the “real” values are likely to sit within. Some rudimentary sensitivity analysis would also be possible.

- The paper would benefit from ensuring that the methods section is reflected in the results section (see my comments, for example about the literature review, which is not really mentioned in the results)

- I think that the discussion section could benefit from being slightly more concise and focussing on how each aspect of the results adds to our knowledge. For me the key questions for the discussion are: How does this change how equids can be considered in Ethiopia, including discussion of previous estimates? What are the implications of the regional differences? How are the financial outputs associated with draft and transport interpreted alongside the market prices for each species, equids as a whole and alongside other livestock species? What do TLUs add to the conversation?

Line by line comments:

Line 46: insert space before “million” (twice). Check 131% number – percentages used in this way may be confusing so could say “more than doubled” or similar

Throughout, check spacing with references (that there is a space inserted before reference bracket)

Line 68: is there a reference for the statement about being comparable to other livestock?

Line 71: Rephrase – do you mean “… female donkey owners indicate that the donkey improved their lives”?

Line 73: Start a new sentence at “In peri-urban….”

Line 76: Poorly captured where? Your previous paragraph described multiple sources that describe the role of working equids. Are these studies poorly executed, or do you mean something else?

Line 79-80: can you clarify what you mean here? Ethiopian notifiable diseases? WOAH listed diseases? WOAH official disease status? These are three different things.

Line 81 onwards: check reference; this does not seem to be a UN document, but it reads as though this is a UN quote

Line 83, 85: try to avoid political geographical categorisations “developing” “sub-Saharan” etc

Line 89: this line is missing a reference

Line 95: Change have for has

Line 99: delete comma

Line 107: expand GBADs acronym. Check sentence structure

Line 113: I think this sentence needs a little more explanation – maybe this comes later in the paper

Throughout paper: Consider when you mean “economic” and when “monetary”: The economic value of an animal is more than its sale value

Methods

Throughout, and relevant to results – the CSA data are a sample – how is this reflected in the results? i.e. I would expect the population densities etc. to be presented as ranges, otherwise a discussion point on the decision on why this was not performed

Line 127: what is a holder?

Line 131: It would be useful to know the use categories that apply to equids

Line 147: suggest “performed” instead of “done”

Paragraph from Line 148: Please provide additional information –

- How many interviews? How were these distributed across demographic and geographical groups?

- How were interviewees selected? What does “conveniently” mean?

- How was consensus reached – how were these results analysed?

You used Google scholar for your literature search – how did you decide how many results to include? What were the criteria for inclusion? How was this information used? The search terms seem to contain duplicates i.e. as “donkey” is searched for, “cart donkey” would automatically be included. Why did you include “value” in “economic value” – for example, the term “economic contribution” would not be included in the search.

Line 174: use sheep and goats, not shoats

Sestion2.2.2 – the first paragraph is results. I would suggest tabulating this information. Line 188 – How did you decide to use this value? Why did you use a single value and not a range?

Line 192: this starts to answer my earlier question – can you explain how “transportation” and “draught” were defined?

Line 195: You earlier said that you asked interviewees about rental value – were these results not used? Which reference was the RV2011 taken from?

Section 2.2.3: which data were used for human populations?

Results:

Line 216: be specific about years i.e. between 20xx and 20xx- in the future it may be difficult for the reader to appreciate “the last 16 years”

Throughout results: please see my earlier comment about using percentages for changes over time.

Throughout: use consistent number of decimal places

Line 223: check that sentences are complete. I don’t think that you need this last sentence. It may be useful to identify these areas on a map

In table 1: How was rate of change calculated (should be in methods). What unit is used?

Figure 2: I am not sure what this shows in addition to table 1, and these are quite hard to read: for example having the key in the same order as the lines would help. Expand SNNP in the legend. Why are only 8 regions included for Donkeys – from table 1, this is the only species for which data are available in all regions, so it is confusing why regions without data are included for horses and mules, or is it that there are no equids in those regions?

Table 2: check headings – what does “households’ own equids” mean? What is the percentage figure? Do you need table 2 and figure 3?

Section 3.2 feels like it repeats some information from line 227-240

Table 4: In the text you describe “asset market value”, in the table “population/stock economic value”. Throughout the paper ensure consistency when describing different values

Figure 4; related to my comment on Figure 2 – areas where there are no data are presented as zero populations. Is this intentional? I would expect them to be marked differently. Check for distortion – maps on the RHS seem to be wider

Line 286: was this biomass figure calculated or identified in the CSA data

Line 289: “The majority of equids….” Check writing in some places for clarity. When you say “This appeared to be”, was this a number that you calculated? Throughout make sure that it is clear which values were calculated in the study, the language in this section could be tightened up, and this would help the reader understand your key points.

Figure 5: I would avoid pie charts in scientific writing like this. I think it would be helpful to show this as a table, with the raw numbers as well as percentages

Table 5: Can you explain the note in the legend? I don’t understand what this is referring to.

There is very limited information about the results of the literature search. How many articles were identified? How many were included in the results? Which were excluded and why?

Did you have regionally disaggregated results for population density for other livestock? This would be interesting as you would be able to see if there were geographical areas where working equids were proportionally more important compared with other species,

Section 3.4: ensure consistency for using equids / equines

Line 316: This first paragraph reads like introduction / conclusion. Why is the last sentence referenced when you have just presented these data.

Line 320: You don’t describe collecting these data about what working equids are used for – more information in the methods is needed

Line 234: As previously; how were these numbers calculated? What was the range of answers?

Line 326: do you mean table 6?

Line 327: change were to was. Check English in this paragraph i.e. “Most of the service value of equine transportation and draft was created by donkeys. Have you defined what you mean by “service value”? It might be worth explained what you mean by presenting this figure as a “gross” figure

Discussion

Line 349: Why did this bias exist?

Line 358: I am not clear if this increase is in line with an increase in all livestock owners – i.e. are you saying that a larger proportion of livestock owning households own working equids/donkeys, or that more households overall all own working equids/donkeys?

Line 372: I am not clear on the meaning of this sentence. It may be helpful to include in the methods some information about how FAOSTAT data are calculated

Paragraph 373: how does this relate to your results? Do you think that the donkey population would be larger without this trade? Were you expecting bigger increases? Or do you think this increase is to fuel the trade in other countries? The most recent Donkey Sanctuary reports cite Ethiopia as supplying the trade – they may have relevant references

Line 400: I thought that the number of holdings had increased? Or is this referring to a specific region/type?

Line 425 onwards: It may be helpful to expand this discussion. You used an estimate for weight based on similar equid systems. Are there any other data that can support this? With your links to Brooke (and other working equids NGOs) are there any other data sources that you could relate to these figures? When referencing the “TLU methods” in line 431 – I thought that you had used liveweights to identify TLU designations for the equine species? It would be helpful to discuss why biomass calculations (rather than just population numbers and monetary value) has been performed

Line 440: Can you describe why you have arrived at this conclusion?

Line 442: Perhaps this needs more explanation?

Line 443: Here when you refer to economic output are you only referring to transportation and draught work?

Line 445: This is the first time that it is mentioned that other authors have calculated equine output. How are the methodologies different? How do you know that the work was underestimated?

Line 451: This support my earlier comments on presenting ranges

Line 456: data “are”

Reviewer #2: this research presents the results of an investigation conducted as a part of the GBAD program, focusing on equids in Ethiopia.

the manuscript is well written, and results are presented with simple figures and tables as summaries.

some minor comments.

I'm wondering whether the section l95-114 are adequately placed in introduction, as they seem to give some rationale for methodlocial choices and background info

the ref 19 is missing in the text, around l 106

l 198: it seems that you could simplify the writing of the equation

l 234: why do you present % in the texte, and not in the table? quite confusing

l 336-344: it s already said in introduction

l 345 346. I could understand the link between the growth of equid and human population, but here you seem to speculate, do you have elements that show this? it is confusing, particularly as reading through the manuscript, we could doubt on the quality of original data used.

major concerns

my major concerns stem from the lack of clarity on:

- the data used and how they were used

- the lack of sensitivity analysis around the final numbers provided

- some assumption s made that are questionable.

section 2.1.1: I ma wondering how do you aggregate the 2 databases that you mention, as they seem to provide different information. how you manage the level of discrepancies between the 2 when it comes to equid population is unclear.

as for the market price, you dot detail if you van time series (which could match the population data)

l 148 onwards. we don't have any specification on how were conducted the interviews, how many.... as the market price appears to be a critical value for you to estimated the economic value, one could expect more details here.

the section 2.2.2. speaks of the exchange value to equids. reading through, we can see huge variation between estimates 157.7 or 445 USD for a multiple fro example. it s about 3 time more. While I can totally understand the difficulty to gather robust estimates, it raises red flags on the validity of the results, which are not weighted in anyway int he rest of the manuscript.

l 196: you consider the inflation rate, but we do not know where you identify the rate.

l 216: I'm questioning the likelihood of the results: an increase of 131% is huge, and it may be linked to at least 3 factors: a "real increase", a reporting bias (change in monitoring system over time for example), a lack of representativity of the sample. you have to reassure he reader on the validity of your findings.

section 3.2

one can see a difference between regions and population density. what we do not know, is the relationship between these 2 factors. I'm wondering if expressing the results in density/km2-ha would not be a way to combine the two, as we can speculate that some areas are of course more dense than others. the question I'm asking is what is the main determinant of equid density? the area or the human population?

how important is to present the results par area?

section 3.3

the economic value is estimated by multiplying the population and individual exchange values (I think). I already raised concerns about the initial estimates. additionally, I'm wondering f it really captures the economic value for many reasons:

first this is a snapshot, likely to be unrealistic if you position these values in a real market. a snapshot evaluation is probably mis-estimating the "true" value, as there will be market adjustments

overall we can speculate the the exchange value reflects the use value (see section 3.4), you coule probably discuss the links between the two.

this section 3.4. is actually interesting (See comment above), but lack of clarity, stemming for the lack of clarity in the methods.

one could put in parallel and discuss how the use and exchange values articulate.

l 434: you present a comparison between the economic values of different species, but we lack the outcomes of the findings.

6. PLOS authors have the option to publish the peer review history of their article (what does this mean?). If published, this will include your full peer review and any attached files.

Reviewer #1: **Yes:**

Reviewer #2: No

---

## [Author Response · Author response to Decision Letter 0]

4 Jul 2023

Response to reviewers 

We would like to say thank you for both reviewers and the editor for their constructive feedback and questions. We have tried to incorporate the feedback and provide clarification response for the questions as follows written in purple.

Thank you!

Response to reviewer #1: Many thanks to the authors for submitting this interesting paper. I have three main comments and some smaller points made by line at the end of this review

- My main question reading this paper is why the results are not presented as ranges. The CSA data sampling methodology are included, and therefore a confidence interval or similar could be calculated. In a similar way, there is insufficient information about how other values are calculated, e.g. from key informant intervals. Presenting these interim figures, as well as the final figures, as ranges, would help the reader understand the range that the “real” values are likely to sit within. Some rudimentary sensitivity analysis would also be possible.

- We have tried to present our result with range value for equid population, number of service days for transportation and market value from the interviewees. We didn’t include range value for equids used for draft service day estimates since we used data from Metaferia et al., (2010) and there was no indication of uncertainty in the research.

- The paper would benefit from ensuring that the methods section is reflected in the results section (see my comments, for example about the literature review, which is not really mentioned in the results)

- We have tried to incorporate the feedback related with methodologies (please see corrected version line 116-175)

- I think that the discussion section could benefit from being slightly more concise and focusing on how each aspect of the results adds to our knowledge. For me the key questions for the discussion are: How does this change how equids can be considered in Ethiopia, including discussion of previous estimates? What are the implications of the regional differences? How are the financial outputs associated with draft and transport interpreted alongside the market prices for each species, equids as a whole and alongside other livestock species? What do TLUs add to the conversation?

- We have tried to make more concise and related with the results of our work and include your comments to the discussion (please see corrected manuscript line 251-334)

Line by line comments:

Line 46: insert space before “million” (twice). Check 131% number – percentages used in this way may be confusing so could say “more than doubled” or similar (corrected line 46 and 47) 

Throughout, check spacing with references (that there is a space inserted before reference bracket)

Checked and addressed, accordingly 

Line 68: is there a reference for the statement about being comparable to other livestock?

The same authors (Admassu and Shiferaw 2011) made this statement 

Line 71: Rephrase – do you mean “… female donkey owners indicate that the donkey improved their lives”?

Addressed, accordingly line 72.

Line 73: Start a new sentence at “In peri-urban….”

Addressed, accordingly line 75

Line 76: Poorly captured where? Your previous paragraph described multiple sources that describe the role of working equids. Are these studies poorly executed, or do you mean something else?

As indicated, there are few studies and most of the papers are reports and bulletins however, they do not adequately address the actual contribution of donkeys to household livelihoods or household income attributable to ownership and donkey use. In addition, despite the fact that donkeys are distributed and used in all parts and production systems of Ethiopia, the indicated studies were conducted around the central part and urban systems. And these studies are not enough to present donkeys' social and economic value compared to their number in the country and their contribution. That is why I wrote as “poorly captured”

Line 79-80: can you clarify what you mean here? Ethiopian notifiable diseases? WOAH listed diseases? WOAH official disease status? These are three different things.

I was meant to say WOAH listed diseases and I made necessary changes please see line 81

Line 81 onwards: check reference; this does not seem to be a UN document, but it reads as though this is a UN quote Corrected line 82-84

Line 83, 85: try to avoid political geographical categorisations “developing” “sub-Saharan” etc

Corrected as low- and middle-income countries' line 84

Line 89: this line is missing a reference - Corrected accordingly 

Line 95: Change have for has - Corrected accordingly

Line 99: delete comma- Corrected accordingly

Line 107: expand GBADs acronym. Check sentence structure-Corrected accordingly

Throughout paper: Consider when you mean “economic” and when “monetary”: The economic value of an animal is more than its sale value

Yes, we agree and well accepted, corrected throughout the paper 

Methods

Throughout, and relevant to results – the CSA data are a sample – how is this reflected in the results? i.e. I would expect the population densities etc. to be presented as ranges, otherwise a discussion point on the decision on why this was not performed

- We have tried to present our result with range value for equid population, number of service days for transportation and market value from the interviewees. We didn’t include range value for draft service day estimates of equids since we used data from Metaferia et al., (2010) and there was no indication of uncertainty in the research.

Line 127: what is a holder?

It was mean to say livestock holder, corrected accordingly line 122

Line 131: It would be useful to know the use categories that apply to equids.

The purpose categories, such as transport, draught, and other uses are incorporated line 127

Line 147: suggest “performed” instead of “done”- corrected accordingly

Paragraph from Line 148: Please provide additional information –

- Because MoTI does not report equids market prices in the system, seventeen (7 knowledgeable elders and 10 livestock market brokers) were interviewed for the current average market price and number of service days for transportation, from 17 districts where Brooke (an international NGO) operates to protect and maintain the health and welfare of equids in Oromia, Amhara, and Southern Nations Nationalities and Peoples (SNNPR) regions. These are dominant areas where donkeys, horses, and mules are used for transportation and draught power. These data are collected by local Brooke representatives and cluster team leaders after receiving oral consent. One interviewee per district were selected purposively based on their knowledge of use frequency and equids' market price. We used questions like "What is the current market price of equids?" In addition, the number of service days worked by equids each week was asked in the interview. (line 130-139)

- How were interviewees selected? What does “conveniently” mean?

Interviewees were selected purposively based on their knowledge of use frequency and equids' market price (line 137). 

- How was consensus reached – how were these results analysed? The mean market price and the number of service days were calculated.

You used Google scholar for your literature search – how did you decide how many results to include? What were the criteria for inclusion? How was this information used? The search terms seem to contain duplicates i.e. as “donkey” is searched for, “cart donkey” would automatically be included. Why did you include “value” in “economic value” – for example, the term “economic contribution” would not be included in the search.

- - We decided not to utilize the data from the lit. review since we did not follow the necessary steps of the systematic literature search, the variation between the paper is vast, we agreed to use primary data (interview) for estimating the monetary value of equids.

Line 174: use sheep and goats, not shoats -Corrected, accordingly (line 126)

Sestion2.2.2 – the first paragraph is results. I would suggest tabulating this information.

The market value is included in result section as per the feedback (line 206-212) 

Line 188 – How did you decide to use this value? Why did you use a single value and not a range?

We used the mean value of market price and number of service days from the interview and decided not to utilize the lit review results 

Line 192: this starts to answer my earlier question – can you explain how “transportation” and “draught” were defined? Addressed as a following: -

The CSA (20) data disaggregated horses, donkeys and mules aged 3 years and above according to their services, mainly as transportation, draught, and others, and this categorization was used to estimate biomass, stock monetary, and service value. Working equids used for transporting goods or people by pulling carts or packing as transportation by the CSA, whilst working equids used for agricultural ploughing are categorized as draught and breeding equids as others (line 154-159). 

Line 195: You earlier said that you asked interviewees about rental value – were these results not used?

Which reference was the RV2011 taken from? Corrected (line 140-146)

The rental value of equids for transportation, as well as rental value and number of service days for draught usage, were obtained from a study done by Metaferia et al (30). By considering the inflation rate

The reference rental value in 2011 was taken from Metaferia et al (30), while the country's average inflation rates were obtained from CSA annual reports.

Section 2.2.3: which data were used for human populations? Addressed as a following:- (line 163-167)

The geographical distribution of equids population was mapped using zone-level number of equids per capita (since the lower scale livestock data available are zone level), calculated as equids population divided by the human population, with human population data obtained from CSA (20). Similarly, equid population per km2 land area was mapped at the zone level taken from MoAG (31).

Results:

Line 216: be specific about years i.e. between 20xx and 20xx- in the future it may be difficult for the reader to appreciate “the last 16 years” -corrected Accordingly – line 178-179

Throughout results: please see my earlier comment about using percentages for changes over time.

Corrected throughout the paper

Throughout: use consistent number of decimal places 

Corrected throughout the paper 

Line 223: check that sentences are complete. I don’t think that you need this last sentence. It may be useful to identify these areas on a map

we agree it already included in the tables and corrected in the document 

In table 1: How was rate of change calculated (should be in methods). What unit is used?

we have changed in to percentage change: To calculate the equid population growth change, we used the difference between the initial equine population in 2004 and the population in 2020, then divided by the initial population and multiplied by 100. And I have incorporated it in the methods (please see table 1)

Figure 2: I am not sure what this shows in addition to table 1, and these are quite hard to read: for example having the key in the same order as the lines would help. Expand SNNP in the legend. Why are only 8 regions included for Donkeys – from table 1, this is the only species for which data are available in all regions, so it is confusing why regions without data are included for horses and mules, or is it that there are no equids in those regions?

Yes, we agree there is no difference between Table 1 and Figure 2, Figure 2 is removed. However, in response to your question, "Why only 8 regions?" some data are lacking in some regions in CSA report and it’s not captured by the graph. 

Table 2: check headings – what does “households’ own equids” mean? – it was meant to say equid holders -corrected (line 204) What is the percentage figure? It is to show the change in number of equine holders between the years. 

Do you need table 2 and figure 3?

Because it has the same messages with table 2 we removed fig 3 

Section 3.2 feels like it repeats some information from line 227-240 Corrected please see 215-220)

Table 4: In the text you describe “asset market value”, in the table “population/stock economic value”.

Corrected accordingly Throughout the paper 

Figure 4; related to my comment on Figure 2 – areas where there are no data are presented as zero populations. Is this intentional? Yes, we tried to present them with the range. I would expect them to be marked differently (we can present differently though). Check for distortion – maps on the RHS seem to be wider please see page 7 amended version. 

Line 286: was this biomass figure calculated or identified in the CSA data

We have calculated the biomass using the CSA population data and the TLU conversion from the literature (please see line 151-135)

Line 289: “The majority of equids….” Check writing in some places for clarity. When you say “This appeared to be”, was this a number that you calculated? Throughout make sure that it is clear which values were calculated in the study, the language in this section could be tightened up, and this would help the reader understand your key points.

we have made necessary changes as per the feedback 

Figure 5: I would avoid pie charts in scientific writing like this. I think it would be helpful to show this as a table, with the raw numbers as well as percentages

was meant to show the difference by comparing with other livestock species of the country. Yes, I accept the feedback and include the point in the text line 228-232

Table 5: Can you explain the note in the legend? I don’t understand what this is referring to.

It was meant to express if there is a variation in number of population of equids in different production system with the total it is because we used the categorization of zone using MoAG and the number is from CSA. 

There is very limited information about the results of the literature search. How many articles were identified? How many were included in the results? Which were excluded and why?

As I indicated above, because we did not follow the proper steps of literature search and because we want relay on the primary data (interview) for estimation of economic value of equids, we decides not to use the data from literature review. 

Did you have regionally disaggregated results for population density for other livestock? This would be interesting as you would be able to see if there were geographical areas where working equids were proportionally more important compared with other species,

I didn’t but I know colleagues in GBADs are working on it.

Section 3.4: ensure consistency for using equids / equines

Line 316: This first paragraph reads like introduction / conclusion. Why is the last sentence referenced when you have just presented these data.

Deleted as per the feedback. 

Line 320: You don’t describe collecting these data about what working equids are used for – more information in the methods is needed

Line 324: As previously; how were these numbers calculated? What was the range of answers?

The mean value is calculated and expressed with range

Line 326: do you mean table 6? Corrected included in the methods with amended version line 128-140

Line 327: change were to was. Check English in this paragraph i.e. “Most of the service value of equine transportation and draft was created by donkeys. Corrected 

Have you defined what you mean by “service value”? It might be worth explained what you mean by presenting this figure as a “gross” figure – “gross” was to say the value from total population but we have now used service value 

Discussion

Line 349: Why did this bias exist?

this could be due to an increase in awareness of owners to report the presence of large numbers of donkeys as livestock to the survey; because donkeys are not food animals in Ethiopia, their owners may not have previously reported them as livestock assets in the survey in the same way that cattle, sheep, or goats are, and this is supported by Starkey and Starkey. Line 258-261

Line 358: I am not clear if this increase is in line with an increase in all livestock owners – i.e. are you saying that a larger proportion of livestock owning households own working equids/donkeys, or that more households overall all own working equids/donkeys? Corrected – it was to say a larger proportion of livestock owning households own working equids

Line 372: I am not clear on the meaning of this sentence. It may be helpful to include in the methods some information about how FAOSTAT data are calculated. It is now corrected- but until 2017 FAOSTAT used to report the data from CSA but after that even though I don’t know the reason starting from 2018 they start to report using imputation. 

Paragraph 373: how does this relate to your results? Do you think that the donkey population would be larger without this trade? Were you expecting bigger increases? Or do you think this increase is to fuel the trade in other countries? The most recent Donkey Sanctuary reports cite Ethiopia as supplying the trade – they may have relevant references- corrected please see 280-286 amended version

Line 400: I thought that the number of holdings had increased? Or is this referring to a specific region/type?

Yes, the number of holdings is increased, but there are some reductions in number of mules holdings 

It’s now corrected.

Line 425 onwards: It may be helpful to expand this discussion. You used an estimate for weight based on similar equid systems. Are there any other data that can support this? With your links to Brooke (and other working equids NGOs) are there any other data sources that you could relate to these figures? When referencing the “TLU methods” in line 431 – I thought that you had used liveweights to identify TLU designations for the equine species? It would be helpful to discuss why biomass calculations (rather than just population numbers and monetary value) has been performed

We have tried but we could not get data related with biomass of equids 

Can you please see line 309-313 amended version 

Line 440: Can you describe why you have arrived at this conclusion? It is now Corrected I was meant to say as the country can benefit with low input 

Line 442: Perhaps this needs more explanation? This is the work that I mentioned which is conducted by other GBADs colleagues 

Line 443: Here when you refer to economic output are you only referring to transportation and draught work? Yes, it the service value of equids it is now corrected 

Line 445: This is the first time that it is mentioned that other authors have calculated equine output. How are the methodologies different? How do you know that the work was underestimated? These are the reports from ministry of trade and finance and IGAD not proper studies their methodologies are not well described, that is why I did not indicate before.

Line 451: This support my earlier comments on presenting ranges corrected most the results are now supported with range 

Line 456: data “are” corrected. 

Response to reviewer #2: 

This research presents the results of an investigation conducted as a part of the GBAD program, focusing on equids in Ethiopia.

the manuscript is well written, and results are presented with simple figures and tables as summaries.

some minor comments.

I'm wondering whether the section l95-114 are adequately placed in introduction, as they seem to give some rationale for methodlocial choices and background info

It is to give some intro about the data I used for the paper, I can still edit as per your feedback 

the ref 19 is missing in the text, around l 106

That is my statement that is why I did not put a reference 

l 198: it seems that you could simplify the writing of the equation

corrected please line 140-146

The rental value of equids for transportation, as well as rental value and number of service days for draught usage, were obtained from a study done by Metaferia et al (30). By considering the inflation rate in the last decade as indicated in the formula below,

RV2021=RV2011+(RV2011*IR2011)+⋯+RV2020+(RV2020*IR2020)

 Where, RV=Rental value, IR= Average inflation rate of the year

The reference rental value in 2011 was taken from Metaferia et al (30), while the country's average inflation rates were obtained from CSA annual reports.

l 234: why do you present % in the text, and not in the table? quite confusing

My mistake corrected accordingly please line 188-194 amended version

l 336-344: it s already said in introduction

Corrected accordingly line 250 amended version 

l 345 346. I could understand the link between the growth of equid and human population, but here you seem to speculate, do you have elements that show this? it is confusing, particularly as reading through the manuscript, we could doubt on the quality of original data used.

corrected please see 253-269

Over the last fifteen years (2004-2020), the equid population has grown by more than double, this is primarily due to increases in the number of holdings, which may be related to the increase in human population. For instance, in rural Ethiopia, where 85% of the population of the country lives, when a new family is established, most begin living by owning a donkey as a working animal for transporting both farm and non-farm items because of poor road-network development. However, this could be due to an increase in awareness of owners to report the presence of large numbers of donkeys as livestock to the survey; because donkeys are not food animals in Ethiopia, their owners may not have previously reported them as livestock assets in the survey in the same way that cattle, sheep, or goats are, and this is supported by Starkey and Starkey (21).

major concerns

my major concerns stem from the lack of clarity on:

- the data used and how they were used

- the lack of sensitivity analysis around the final numbers provided

- some assumption s made that are questionable.

section 2.1.1: I ma wondering how do you aggregate the 2 databases that you mention, as they seem to provide different information. how you manage the level of discrepancies between the 2 when it comes to equid population is unclear.

I was using both CSA and FAOSTAT to show the trend difference compared to CSA.

I corrected it please see line 121-122

as for the market price, you dot detail if you van time series (which could match the population data)

I wish I could show the trend with the market price as well but the market price data are lacking/not sufficient that is why we use the data from the interview please see line 132-140

l 148 onwards. we don't have any specification on how were conducted the interviews, how many.... as the market price appears to be a critical value for you to estimated the economic value, one could expect more details here.

my mistake, corrected please see line 132-140

Because MoTI does not report equids market prices in the system, seventeen (7 knowledgeable elders and 10 livestock market brokers) were interviewed for the current average market price and number of service days for transportation, from 17 districts where Brooke (an international NGO) operates to protect and maintain the health and welfare of equids in Oromia, Amhara, and Southern Nations Nationalities and Peoples (SNNPR) regions. These are dominant areas where donkeys, horses, and mules are used for transportation and draught power. These data are collected by local Brooke representatives and cluster team leaders after receiving oral consent. One interviewee per district were selected purposively based on their knowledge of use frequency and equids' market price. We used questions like "What is the current market price of equids?" In addition, the number of service days worked by equids each week was asked in the interview.

the section 2.2.2. speaks of the exchange value to equids. reading through, we can see huge variation between estimates 157.7 or 445 USD for a multiple fro example. it s about 3 time more. While I can totally understand the difficulty to gather robust estimates, it raises red flags on the validity of the results, which are not weighted in anyway in the rest of the manuscript.

- We decided not to utilize the data from the lit. review since we did not follow the necessary steps of the systematic literature search, the variation between the paper is vast, we agreed to use primary data (interview) for estimating the monetary value of equids.

l 196: you consider the inflation rate, but we do not know where you identify the rate.

corrected please see line 147-148 � The reference rental value in 2011 was taken from Metaferia et al (26), while the country's average inflation rates were obtained from CSA annual reports.

l 216: I'm questioning the likelihood of the results: an increase of 131% is huge, and it may be linked to at least 3 factors: a "real increase", a reporting bias (change in monitoring system over time for example), a lack of representativity of the sample. you have to reassure the reader on the validity of your findings.

I have tried to discuss the possible reasons can you please see line 255-271 for me it can be both the real increase and also reporting bias as there are unrealistic surges are there in the trend and I mentioned this in the discussion 

section 3.2

one can see a difference between regions and population density. what we do not know, is the relationship between these 2 factors. I'm wondering if expressing the results in density/km2-ha would not be a way to combine the two, as we can speculate that some areas are of course denser than others. the question I'm asking is what is the main determinant of equid density? the area or the human population?

how important is to present the results par area?

The main determinant for equid density is human population and we have made necessary changes as per your question and feedback please see line 216-229. 

section 3.3

the economic value is estimated by multiplying the population and individual exchange values (I think). I already raised concerns about the initial estimates. additionally, I'm wondering f it really captures the economic value for many reasons:

first this is a snapshot, likely to be unrealistic if you position these values in a real market. a snapshot evaluation is probably mis-estimating the "true" value, as there will be market adjustments

overall we can speculate the the exchange value reflects the use value (see section 3.4), you coule probably discuss the links between the two.

I have changed the word economic value to monetary value as it is more general and it describe our result better and we have indicated the result with range value. Please see line 227-247.

this section 3.4. is actually interesting (See comment above), but lack of clarity, stemming for the lack of clarity in the methods. Corrected, I have now included in the method section please see line 150-153

one could put in parallel and discuss how the use and exchange values articulate.

l 434: you present a comparison between the economic values of different species, but we lack the outcomes of the findings. Corrected as per the feedback, I was trying to compare and show the difference in biomass and monetary value between species by considering the data from GBADs colleagues, but I now removed the point because I didn’t touch in my result and even methods. 

Response for Editorial comments 

1. We follow PLOS one authors guidelines to meet editorial requirement. 

2. We removed this reference because we don’t use it anymore. 

4. The map is our work and we have included it in the method section please see line 163-167

The geographical distribution of equids population was mapped using zone-level number of equids per capita (since the lower scale livestock data available are zone level), calculated as equids population divided by the human population, with human population data obtained from CSA (20). Quantum GIS 3.16.11 software (QGIS Geographic Information System. Open-Source Geospatial Foundation Project. http://qgis.osgeo.org ) was used to map equid populations.

I thank you both reviewers and the editor, this will strengthen our paper.

Respectfully, Girma

---

## [Decision Letter · Decision Letter 1]

6 Oct 2023

PONE-D-23-03880R1Population, distribution, biomass, and economic value of equids in EthiopiaPLOS ONE

Dear Dr. Asteraye,

Thank you for submitting your manuscript to PLOS ONE. After careful consideration, we feel that it has merit but does not fully meet PLOS ONE’s publication criteria as it currently stands. Therefore, we invite you to submit a revised version of the manuscript that addresses the points raised during the review process.

We look forward to receiving your revised manuscript.

Kind regards,

Chisoni Mumba

Academic Editor

PLOS ONE

Journal Requirements:

Additional Editor Comments:

Reviewer 1 still has a few comments mainly because he did not see the copy with tracked changes. Please make any minor revisions from Reviewer 1 and resubmit within a fourtnight.

Reviewers' comments:

Reviewer's Responses to Questions

**Comments to the Author**

1. If the authors have adequately addressed your comments raised in a previous round of review and you feel that this manuscript is now acceptable for publication, you may indicate that here to bypass the “Comments to the Author” section, enter your conflict of interest statement in the “Confidential to Editor” section, and submit your "Accept" recommendation.

Reviewer #1: (No Response)

Reviewer #3: All comments have been addressed

2. Is the manuscript technically sound, and do the data support the conclusions?

Reviewer #1: Yes

Reviewer #3: Yes

3. Has the statistical analysis been performed appropriately and rigorously? 

Reviewer #1: No

Reviewer #3: Yes

4. Have the authors made all data underlying the findings in their manuscript fully available?

Reviewer #1: Yes

Reviewer #3: Yes

5. Is the manuscript presented in an intelligible fashion and written in standard English?

Reviewer #1: Yes

Reviewer #3: Yes

6. Review Comments to the Author

Reviewer #1: Thank you for providing this improved manuscript. Please note that it would be helpful to provide a version with marked up tracked changes that show where changes have been made to the document, not wholesale deletions and reinsertions. Additionally it would be helpful if the responses consistently contained line numbers that refer to the most recent version of the manuscript. It is very time consuming to re-review a document presented in this way. The document I have access to has four different versions of the paper -I have based this review on the one starting on page 68. This means I may miss some changes, as I cannot reference them by line number, or comment on parts of the manuscript that are unchanged.

Generally, two main comments still require some changes. It is good to see that the suggestion about including a numerical range to represent the sampling uncertainty. But what does the ranges presented represent and how were they calculated? – confidence intervals? Standard deviation? These should inform your sensitivity analysis, which in insufficiently described in the methods, results and discussion.

Some additional points

Line 68: In this case “demonstrated that equids provide 14% of annual household income … which was comparable to other livestock species in that study”.

Line 77: This still feels like a contradiction to me and would benefit from some critical thinking (with reference to line 90-91 as well)

Line 81: The manuscript does not reflect the response to reviewers

Line 173: suggest: “The equid population percentage change was calculated by dividing the difference between the initial equine population in 2004 and the population in 2020 by the initial population and multiplying by 100”

Table 2: explain what these numbers are a percentage of – all equid holders or all livestock holders. Make sure that the title represents the tables contents – it does not show the change, simply the populations in the two years

Table 4 appears to repeat table 3 (although the numbers are slightly different). Can you include a clear explanation?

Line 212: “Because the data is insufficient, we did not estimate the effective service days of

equids for draught usage.” – but then this seems to have been done in table 5

Line 267: Which scientific growth rate estimates are you referring to?

Line 315 onwards: I think it would still be beneficial to discuss these other estimates in more depth, even if they do not have clearly defined methodologies (even including in the manuscript that the methodologies are not clearly defined.)

The manuscript would benefit from a thorough proof read: e.g. consistency of decimal places, some verb agreements, use of data as either a singular or plural throughout, not a mix

Reviewer #3: (No Response)

7. PLOS authors have the option to publish the peer review history of their article (what does this mean?). If published, this will include your full peer review and any attached files.

Reviewer #1: No

Reviewer #3: No

---

## [Author Response · Author response to Decision Letter 1]

10 Nov 2023

Dear reviewer-1: Please accept my apologies for making the review challenging; I will work properly after this. Dear editor, apologise for not returning my response within fortnights, I was out of the city for data collection, and the internet connectivity was very limited, which made it difficult for me to send my response in time. I appreciate your patience and understanding in this regard. Thank you.

Please refer to the indicated line numbers in the recent version of the manuscript and here in this box I have included your questions with my responses. 

Generally, two main comments still require some changes. It is good to see that the suggestion about including a numerical range to represent the sampling uncertainty. But,

• What does the ranges presented represent and how were they calculated? – confidence intervals? Standard deviation? These should inform your sensitivity analysis, which in insufficiently described in the methods, results and discussion.

Okay, the range shown is a 95% confidence interval (CI); because the CSA estimates livestock population and indicates a standard error, we calculated the confidence interval to take into account the uncertainty associated with sampling variability; we have calculated using the formula 

 Confidence Interval = The equids population mean ± (Z * Standard Error)

 - Mean: The equids population mean 

 - Z: ≈ 1.96 as CSA calculated the CI at 95%

 - Standard Error: the SE value from CSA 

The sampling uncertainty was expressed by calculating a confidence interval from CSA livestock population estimates and standard errors. This is now included in the method section line 173-74.

Some additional points

Line 68: In this case “demonstrated that equids provide 14% of annual household income … which was comparable to other livestock species in that study”.

Yes, as per the result of Admassu and Shiferaw (2011), working equids contribute 14% of household income and which was comparable to other livestock species. I accept the comment - line 69

Line 77: This still feels like a contradiction to me and would benefit from some critical thinking (with reference to line 90-91 as well)

I have incorporated necessary change for the sentence as per the comment received - line 81-82

Line 81: The manuscript does not reflect the response to reviewers

I have made changes to the paragraph ( line 79 to 83) based on the feedback I received.

Line 173: suggest: “The equid population percentage change was calculated by dividing the difference between the initial equine population in 2004 and the population in 2020 by the initial population and multiplying by 100”

Thank you and I accept your suggestion fully, and addressed, accordingly, please see line (170-72)

Table 2: explain what these numbers are a percentage of – all equid holders or all livestock holders. Make sure that the title represents the tables contents – it does not show the change, simply the populations in the two years

This proportion was determined based on total livestock holdings, and I have updated the title to reflect this (line 203-4).

Table 4 appears to repeat table 3 (although the numbers are slightly different). Can you include a clear explanation?

As mentioned in the methodology section, the CSA separated the equine population based on species, sex, and purpose. This information was used to calculate the biomass and the monetary value of the stock. The slight difference in the number between the total (Table 3) and the categorized (Table 4) is a result of the disaggregation process. I have included the explanation for the difference, please see line 237-8. 

Line 212: “Because the data is insufficient, we did not estimate the effective service days of equids for draught usage.” – but then this seems to have been done in table 5

As it indicated in method section (line 140-42) rental value and number of service days for draft usage, were obtained from a study done by Metaferia et al (2010) for calculation to estimates of service value of equids. Therefore, the draught usage service days indicated in table 5 are from Metaferia et al (2010). Kindly, If I am expected to write under a table 5 as a note, I can do that. 

Line 269: Which scientific growth rate estimates are you referring to?

I should not have used the word "scientific" in that sentence. I was referring to the unexpected growth of the equid population in 2006 with 19.2%, which is quite different from the rest of the consecutive years, and if the equid birth rate is one foal per year (which is equivalent to a 100% foaling rate), this would not result in the type of growth CSA indicated in 2006. Please see (lines 274 and 75) for a correction.

Line 315 onwards: I think it would still be beneficial to discuss these other estimates in more depth, even if they do not have clearly defined methodologies (even including in the manuscript that the methodologies are not clearly defined.)

I have tried to include their methods and discuss with our result, can you please see line 321-330

The manuscript would benefit from a thorough proofread: e.g. consistency of decimal places, some verb agreements, use of data as either a singular or plural throughout, not a mix- I attempted to make decimal places consistent manly in the tables by approximating.

---

## [Editor Report · Decision Letter 2]

22 Nov 2023

Population, distribution, biomass, and economic value of equids in Ethiopia

PONE-D-23-03880R2

Dear Dr. Asteraye,

We’re pleased to inform you that your manuscript has been judged scientifically suitable for publication and will be formally accepted for publication once it meets all outstanding technical requirements.

Kind regards,

Chisoni Mumba

Academic Editor

PLOS ONE
---

## [Editor Report · Acceptance letter]

10 Jan 2024

PONE-D-23-03880R2 

PLOS ONE

Dear Dr. Asteraye, 

I'm pleased to inform you that your manuscript has been deemed suitable for publication in PLOS ONE. Congratulations! Your manuscript is now being handed over to our production team.

Kind regards, 

on behalf of

Dr Chisoni Mumba 

Academic Editor

PLOS ONE